# Mindfulness-Based Interventions for the Treatment of Aberrant Interoceptive Processing in Substance Use Disorders

**DOI:** 10.3390/brainsci12020279

**Published:** 2022-02-17

**Authors:** April C. May, Chrysantha Davis, Namik Kirlic, Jennifer L. Stewart

**Affiliations:** 1San Diego State University/University of California, San Diego Joint Doctoral Program in Clinical Psychology, San Diego, CA 92037, USA; 2Laureate Institute for Brain Research, Tulsa, OK 74136, USA; cdavis@laureateinstitute.org (C.D.); nkirlic@laureateinstitute.org (N.K.); jstewart@laureateinstitute.org (J.L.S.); 3Department of Community Medicine, University of Tulsa, Tulsa, OK 74104, USA

**Keywords:** interoception, mindfulness, substance use disorder, insula

## Abstract

Altered interoception, or the processing of bodily signals, has been argued to play a role in the development and maintenance of substance use disorders (SUD). Therefore, interoceptive interventions focusing on bodily awareness, such as mindfulness meditation, may improve treatment outcomes for individuals with SUD. Here we review: (1) subjective, behavioral and brain evidence for altered interoceptive processing in SUD, focusing on insular and anterior cingulate cortices (INS, ACC), key regions for interoceptive processing; (2) research highlighting links between mindfulness and brain function; and (3) extant brain research investigating mindfulness-based interventions in SUD. SUD tend to be characterized by heightened INS and ACC responses to drug cues but blunted interoceptive awareness and attenuated INS and ACC responses during tasks involving bodily attention and/or perturbations. In contrast, mindfulness interventions in healthy individuals are linked to enhanced INS and ACC responses and heightened interoceptive awareness. It is crucial for future research to identify: (1) whether mindfulness-based treatments are efficacious across substance classes; (2) what particular approaches and dosages show the largest effect sizes in enhancing INS and ACC function to non-drug stimuli and reducing responsivity to substance cues, thereby improving SUD treatment outcomes (reducing drug craving and relapse).

## 1. Impact and Treatment of Substance Use Disorders (SUDs)

In 2019, a reported 20.4 million United States citizens (approximately 6% of the total population) aged 12 or older were diagnosed with at least one substance use disorder (SUD) [1]. SUD are associated with negative consequences for the individual including poor physical and mental health, interpersonal difficulties, and heightened mortality risk [2,3,4]. High rates of substance use also pose significant societal issues including increased rates of crime and incarceration and substantial healthcare resource burden [5]. Despite the elevated rates of SUD, effective intervention options are lacking and rates of relapse following treatment are egregious, with up to 50% of patients relapsing within 6 months [6], thus warranting the need for improved treatments. We argue that the development of effective interventions should be informed by our growing understanding of how brain functions are altered among individuals with SUD (iSUD). Specifically, a growing body of literature suggests that altered processing of bodily signals, known as interoception, plays a key role in the development and maintenance of SUD [7,8] and as such, interoception is a prime target for intervention with techniques focusing on bodily awareness. These techniques, such as mindfulness meditation or mindfulness-based interventions (MBI), have been demonstrated to alter brain activity among non-substance using individuals, and therefore hold promise for modifying aberrant brain activity observed within iSUD [8,9]. Given this, below we review: (1) functional magnetic resonance imaging (fMRI) evidence of altered interoceptive processing among iSUD; (2) research highlighting links between mindfulness and brain function in non-substance-using individuals; and (3) the limited published neuroimaging research examining mindfulness-based interventions among iSUD. Lastly, directions for future research are discussed.

## 2. Interoception

Interoception refers to the process of sensing the internal physiological state of the body by receiving and integrating body-relevant signals with external stimuli [10]. Through this process, an individual develops a mental representation of the overall state of the body comprising a wide range of physiological sensations such as cardiovascular arousal, respiration, pain, temperature, itch, hunger, and thirst [11]. Interoception is conceptualized as a homeostatic process, such that an individual’s sensing of this body-related information motivates the decision to approach or avoid certain stimuli in the environment with the goal of maintaining or restoring equilibrium. In the healthy individual, this may lead to seeking out sustenance when hungry, or sleeping when tired; however, aberrant interoceptive processes have been implicated in various psychiatric disorders. For example, because interoceptive feedback contributes to other cognitive processes including emotion regulation, decision-making, reward processing, and cognitive control [7], alterations in interoception are associated with mood, anxiety, and addictive disorders [12,13,14].

The insular cortex (INS) has been characterized as the ‘hub’ of interoception [15]. The posterior INS receives and processes interoceptive sensory information delivered by the thalamus. This information is passed onto the anterior INS where it is integrated with higher order cognitive and emotional information. INS activation is regularly observed to coincide with activation in the anterior cingulate cortex (ACC), suggesting that the ACC plays an integral role in the integration of interoceptive information as well [16,17]. Other adjacent structures, including the inferior frontal operculum, inferior frontal gyrus (IFG), amygdala (for integration of emotionally salient information), and somatomotor cortex are jointly activated with INS when attending to interoceptive events [16,17]. Alterations in this neural network may be reflected by variations in interoceptive accuracy (IA) measured as one’s ability to accurately perceive interoceptive signals within the body. IA is typically assessed using heartbeat detection tasks which allow for the objective quantification of individual differences in perceived vs. actual heart beats, with a greater discrepancy indicating reduced interoceptive accuracy [18]. Such alterations may contribute to the development and maintenance of various psychiatric disorders characterized by disruptions in processes dependent on interoceptive input (e.g., emotion regulation, decision-making) [19,20,21].

Individual differences in IA are suggested to be important for how one experiences and responds to internal sensations, with greater IA linked to heightened intensity of physical and emotional sensations [22]. Heartbeat detection paradigms are commonly employed to examine differences in interoceptive processing, as these tasks have been demonstrated to activate regions central to the interoceptive network including bilateral INS, somatomotor, and cingulate cortices, middle frontal gyrus, and amygdala [11,17]. Such tasks have demonstrated an association between IA and various psychiatric disorders. For example, INS activation while attending to one’s heartbeat is positively associated with heartbeat perception accuracy and self-reported anxiety [23]. This association may provide insight to the psychopathology cycle underlying panic attacks, such that individuals with elevated levels of anxiety may be more sensitive to increases in heart rate and perceive such increases as a signal of increased threat. Similarly, research suggests interoceptive processes are altered in iSUD. In these individuals, heightened or blunted bodily signaling may then contribute to substance use, motivating an individual to seek out substances. This, in turn, contributes to the development, maintenance, and recurrence of disordered use.

## 3. The Role of Interoception in SUD

Interoception is thought to provide information that influences an individual’s decision to approach or avoid stimuli in the environment with the aim of achieving equilibrium. Dysfunction in interoceptive brain regions may provide a blunted experience of bodily-relevant signals, leading to an inaccurate representation of the bodily state, referred to as a body prediction error [24,25,26,27]. Consideration of this inaccurate representation during decision-making and emotion regulation processes may lead to maladaptive behavioral adjustments [28], such as seeking stronger rewarding stimuli (drugs such as cocaine and methamphetamine) to register bodily sensations in place of natural rewards. Over time, blunted activity in interoceptive regions (e.g., INS, ACC) paired with exaggerated response to substance-related cues in these same regions within iSUD may lead to an increased behavioral approach towards substances of abuse, thereby perpetuating the cycle of substance use. Alternatively, exaggerated interoceptive responses to negative or uncomfortable stimuli (e.g., increased heart rate) may lead an individual to seek out substances such as alcohol, opioids, or benzodiazepines to alleviate these sensations. Interoception is also linked to drug craving, wherein the anterior INS directs attention away from cognitive tasks and towards interoceptive stimuli, such as the bodily effects of substance use, and cues cravings to use in the future [29]. Below, we review the literature for evidence of interoceptive dysfunction in iSUD.

### 3.1. Altered Interoceptive Processing in Individuals with SUD (iSUD)

#### 3.1.1. Alcohol

Reduced IA has been demonstrated among individuals meeting criteria for an alcohol use disorder (iAUD) using a behavioral heartbeat perception test. The Schandry task is a widely used mental tracking paradigm in which participants are instructed to silently count their heartbeats across trials of varying lengths and IA is calculated using the participant’s reported versus actual number of heartbeats recorded using electrocardiogram. Of note, the validity of the Schandry task has recently been called into question (see [30] for more details). Using this task, recently abstinent iAUD exhibit lower IA than healthy controls (CTL), with lower IA linked to greater self-reported alcohol craving [31]. Jakubczyk and colleagues [32] replicated and extended this finding, showing that iAUD have both lower IA and higher emotional dysregulation (measured by self-report) than CTL. Furthermore, lower IA in iAUD was associated with poorer self-reported ability to accept emotional distress. Within iAUD, poor awareness or ability to identify emotional states has been identified as a risk factor for future alcohol use; it is argued that as these individuals are unable to accurately identify their emotional experiences and cope effectively, they instead turn to alcohol for relief from bodily sensations [33]. This finding suggests that improving IA and, therefore, one’s ability to identify and describe somatic states, may reduce risk of using alcohol consumption as a coping mechanism [32].

The cold pressor task (CPT) is an additional behavioral paradigm used to assess interoceptive responsivity to a physical stressor in which participants are asked to submerge their hand in a cold-water bath for an extended period [34]. This task was used to examine the relationship between stress reactivity and craving and prospective alcohol use among iAUD with and without post-traumatic stress disorder (PTSD). For iAUD only, change in self-reported alcohol craving from pre- to post-CPT predicted magnitude of alcohol use in the month following participation. Additionally, individuals with a higher craving but blunted adrenocorticotrophin (a hormone produced by the pituitary gland in reaction to biological stress) response to the CPT reported significantly greater frequency and intensity of alcohol use during the following month [34]. These findings were interpreted to suggest that individuals with dysregulated biological stress and interoceptive systems may be at the greatest risk for relapse. 

Alterations in interoceptive processing have also been examined among individuals who use alcohol recreationally and iAUD using neuroimaging techniques. Alcohol use among first-year college students was monitored over the course of a year and progression from moderate to heavy drinking, and increased frequency of alcohol-related problems including experiencing hangovers or memory black-outs, feeling sick, missing class, or impaired performance at school, were linked to baseline hyperactivation in INS, ACC, caudate, and orbitofrontal cortex activation to alcohol images [35]. Thus, heightened responsivity to substances cues in interoceptive brain regions prior to AUD may serve as a marker of increased risk for maladaptive drinking.

Moreover, INS activity was identified as potentially contributing to continuance of heavy drinking among non-treatment seeking iAUD [36]. Greater functional connectivity (i.e., temporal correlation of signal changes between spatially distant brain regions) between INS and the hippocampus (implicated in memory formation) and orbitofrontal cortex (reward valuation) in a resting state was observed in iAUD but not CTL. Altered connectivity between these regions and INS may suggest a stronger and more positive representation of the effects of alcohol intoxication in memory that may contribute to sustained alcohol use [36]. These results contrast with findings of reduced functional connectivity at rest in INS, precuneus (implicated in memory and cue reactivity), and postcentral gyrus (primary somatosensory cortex) among individuals deemed to be “drinkers” based on their scores on the Alcohol Use Disorder Identification Test (AUDIT) using a threshold previously demonstrated to detect hazardous drinking [37]. These discrepant findings may be a function of the sample characteristics, suggesting varying patterns of resting INS connectivity among iAUD compared to subthreshold hazardous drinkers.

#### 3.1.2. Stimulants

Interoceptive processing has also been examined among individuals who use stimulants such as methamphetamine and cocaine. Discrepant patterns of brain activation in response to a variety of interoceptive stimuli (e.g., heartbeat sensations, physical soft touch) have been observed among these individuals in comparison to CTL. Using a visceral interoceptive awareness task in which participants were instructed to attend to heartbeat sensations, stomach sensations, or an external visual cue, individuals with methamphetamine use disorder (iMUD) showed blunted anterior/middle INS activation during interoceptive (heartbeat and stomach) trials relative to CTL [8]. Despite this demonstration of a reduced neural response to the interoceptive stimuli, iMUD rated the heartbeat-related sensations as more intense than CTL. Furthermore, iMUD with greater cumulative lifetime stimulant use exhibited greater INS activation to heart sensations, meaning that their brain response approximated that of CTL when compared to iMUD with lower lifetime use. Finally, iMUD with more recent stimulant use reported the greatest intensity of heartbeat sensations. However, within iMUD, INS activation to interoceptive sensations did not significantly correlate with self-reported intensity of these sensations, suggestive of a disconnect between channels of bodily processes. Taken together, findings indicate that recency and chronicity of stimulant use moderate the degree of response to internal signals. Alternatively, individuals meeting criteria for cocaine use disorder demonstrated increased IA during a heartbeat detection task compared to CTL, and IA was inversely related to connectivity between INS and other regions of the interoceptive network including postcentral and temporal regions [38]. Overall, these results suggest a hyperresponsivity of the interoceptive system among individuals with sustained cocaine use. 

Chronic iMUD abstinent for approximately 1 month also show lower INS, thalamus, and striatum responses than CTL to positive gentle tactile stimulation of mechano-receptive c-fibers, which project to anterior INS via the thalamus for interoceptive processing, found in palm and forearm skin [24]. In a related way, iMUD appear to exhibit a diminished neural response when faced with a negative interoceptive stimulus. Stewart and colleagues [39] paired an aversive breathing load with a two-choice prediction task to examine how negative sensations affect decision-making among iMUD as a proxy for making substance use related decisions while not feeling well (e.g., during craving or withdrawal). Across breathing load trials, chronic iMUD exhibited lower posterior INS and ACC activation than CTL the aversive stimulus, as well as ACC when the aversive stimulus was paired with negative feedback. It is important to note that anterior INS activation was lower in iMUD than CTL across all trials, regardless of condition, suggesting an overall blunted INS response that could be attributed to overall differences in attentional resources allocated to the decision-making task. Together, these studies demonstrate a pattern of altered interoception among individuals with years of substance use who meet diagnostic criteria for a use disorder. However, individuals with years of disordered use may differ from individuals with diagnostically subthreshold use in important ways. 

Responsivity to pleasant and aversive interoceptive stimuli has also been examined among individuals who use stimulants (e.g., dextroamphetamine, cocaine, methylphenidate) recreationally, as a predictor of transition to problematic substance use (PSU: 2+ DSM-IV criteria of abuse and/or dependence) [40,41]. Young adults who used stimulants recreationally were followed up over a 3-year period to identify those who transitioned to PSU versus those who discontinued or desisted using stimulants during that period (DSU). PSU, DSU, and a sample of age-matched CTL completed a two-choice prediction task paired with an aversive inspiratory breathing load [41] and a continuous performance task paired with a pleasant soft touch stimulus [40] during fMRI recording to determine whether recreational use or more severe problematic substance use was linked to interoceptive brain dysfunction. Findings did not reveal any group differences in INS activation during the breathing load task, however, PSU, compared to DSU and CTL, exhibited reduced IFG and ACC, and both PSU and DSU, compared to CTL, showed lower thalamus activation during the aversive breathing load [41]. During the soft touch task, PSU exhibited higher anterior INS, IFG, and superior frontal gyrus across trials than DSU or CTL. Furthermore, PSU showed greater middle INS activation than DSU and CTL specifically in response to the soft touch [40]. Overall, these findings suggest that (1) frontocingulate attenuation in the presence of aversive interoceptive perturbations may be an indicator of susceptibility to PSU; and (2) hyperresponsivity in interoceptive regions marks the recent transition to problematic drug use, as opposed to attenuation indexing individuals with chronic use. 

Other investigations have reported reduced INS activation among individuals meeting DSM-IV criteria for methamphetamine abuse, compared to CTL, while viewing threatening or fearful images [42]. Although these findings appear to suggest that individuals who abuse methamphetamine are more like iMUD than PSU, these differences may actually be a function of the type of interoceptive stimuli experienced (emotional vs. physical). Additionally, this finding may be related to differences between diagnostic classification systems used between studies as endorsement of two criteria met requirements for a diagnosis of substance abuse in DSM-IV, but mild SUD in DSM-5. Regardless of this, these findings highlight altered interoceptive processing among individuals with varying levels of use and related problems. Future studies will be useful in clarifying differences in interoceptive processing in accordance with progression of use.

#### 3.1.3. Nicotine

Among individuals who use nicotine, interoception has predominantly been examined in relation to cue-induced craving and relapse. Recently abstinent individuals with nicotine dependence show increased anterior INS functional connectivity to clusters including the precuneus and angular gyrus, regions involved in the default mode network, while viewing smoking cues compared to neutral cues [43]. Previous research has suggested that the anterior INS is responsible for toggling between the default mode network when at rest, and the executive control network when confronted with a task requiring attention [44]. In relation to nicotine use, these findings suggest that heightened connections between anterior INS and precuneus may represent coordinated processing of negative bodily sensations (i.e., craving) and conscious evaluation of the salience of the smoking cues [43]. The magnitude of functional connectivity between these regions was positively correlated with the magnitude of self-report craving, further supporting this interpretation. Relatedly, posterior INS resting-state functional connectivity also appears to play a role in nicotine use, significantly differing between individuals who relapse after an attempt to quit and those who do not; individuals who do not relapse after quitting exhibit greater posterior INS connectivity with pre- and postcentral gyri, primary sensorimotor cortices than those who relapse [45]. These findings were interpreted as suggesting an improved ability to inhibit a motor response to engage in substance use when faced with nicotine craving, resulting in an increased likelihood of cessation.

Comparable findings have been found within female-only samples as well. Females meeting criteria for DSM-IV nicotine dependence demonstrated positive correlations between attentional bias to smoking images and brain activation in INS, amygdala, and hippocampus, potentially signaling an interoceptive representation of smoking and induced craving [46]. In line with previously discussed findings [43], these results suggest that cigarette-smoking individuals with elevated attentional biases to substance cues may more readily shift attention away from other stimuli in favor of smoking-related stimuli, thereby increasing risk of relapse. A follow-up study examined these neural findings in relation to future quit attempts and found that increased attentional bias and INS and ACC activity in response to smoking-related images was associated with ‘slips’, defined as smoking one cigarette after attaining abstinence [46]. Individuals who smoked a cigarette after abstinence also had decreased functional connectivity between INS and ACC/PFC while viewing smoking cue images compared to neutral images. Furthermore, a discriminant analysis that included accuracy and reaction time on an emotional Stroop task and INS/ACC activation, predicted attempt to quit outcomes with 79% accuracy.

Interoception has also been examined among individuals who smoke nicotine using behavioral measures such as a heartbeat tracking and heartbeat discrimination task. Using these tasks, adult non-smokers demonstrated increased IA on the heartbeat-tracking task compared to individuals who reported smoking cigarettes daily for at least 1 year [47]. However, there was no other evidence of interoceptive differences between groups behaviorally or on self-report measures.

Overall, findings from neuroimaging studies provide strong support for altered interoceptive processing among individuals who use nicotine and suggest altered connections between interoceptive regions and regions involved in decision-making, behavioral responses, and emotional processing. Findings from behavioral and self-report data are less robust.

#### 3.1.4. Opioids

Although fewer studies have examined interoception among individuals with opioid use disorder (iOUD), existing research suggests a relationship between brain activity in interoceptive regions and cue-reactivity, craving, and relapse. For instance, iOUD undergoing methadone maintenance treatment (MMT), regardless of dosage, demonstrated reductions in INS, amygdala, and hippocampus, but not OFC or ACC, responsivity to heroin cues (compared to neutral cues) post-methadone administration [48]. These findings may point to a mechanism of efficacy for MMT as it reduced exaggerated responsivity to salient drug cues in brain regions known to be implicated in addiction (e.g., INS, amygdala) while also pointing to a potential weakness of MMT as activation in OFC and ACC were not altered with dosage. These reductions in brain activity following methadone dosage also lasted less than the 24 h interdose interval. In a related way, cue-induced neural activity differs as a function of abstinence duration among iOUD [49]. Deactivation in INS, thalamus, striatum, and posterior cingulate while viewing heroin-related cue images was observed among long-term (over 1 year) abstinent iOUD compared to increased activation in ACC, thalamus, and hippocampus among short-term (approximately 1 month) abstinent iOUD. Moreover, iOUD show lower functional connectivity, a pattern associated with greater relapse frequency over a 26-month follow-up period. Finally, recent work indicates iOUD exhibit lower INS activation when attending to stomach sensations than CTL during a visceral interoceptive awareness task; however, iOUD did not differ from a group of individuals with stimulant use disorders in their stomach-related INS responses, suggesting that blunted interoceptive processing to gastrointestinal sensations is not specific to problems with opioids [8]. Although iOUD did not differ from CTL in INS activation to heart or visual sensations or the self-reported intensity of these sensations, these null findings may be related to the relatively small sample size for the iOUD group. Taken together, these findings suggest that: (1) functioning of various brain regions implicated in interoception including INS, amygdala and thalamus, among others, is altered in iOUD; and (2) interventions that aim to manipulate activity in these regions may be effective in decreasing salience of substance-related cues, extending abstinence, and reducing relapse rates.

#### 3.1.5. Cannabis

Individuals who use cannabis chronically or meet criteria for cannabis use disorder (iCUD) also demonstrate altered functioning in interoceptive regions, however, these alterations appear to differ from the patterns seen in individuals with other SUDs. In contrast to iOUD, iCUD exhibit increased responsivity to cannabis-related stimuli in anterior INS, striatum, and amygdala, and INS activity positively correlates with cannabis craving [50]. Individuals with chronic patterns of cannabis use also demonstrate alterations in INS functional connectivity. Specifically, increased resting-state connectivity within anterior INS but decreased activity between INS and ACC/thalamus is observed among young adult male cannabis users, and these effects appeared to persist after 1 month of abstinence [51]. Altered INS activity is also linked to poor error-monitoring in chronic cannabis users [52]. When completing a response inhibition task, individuals with chronic cannabis use demonstrated lower INS and ACC activation than CTL, although they committed inhibitory control errors at similar rates [52]. Furthermore, within users, lower INS activation is related to higher levels of recent cannabis use. Overall, these findings suggest that individuals who chronically use cannabis have altered interoceptive awareness, including awareness of errors, and may provide support for the theory of addiction that posits that individuals use substances to enhance visceral sensations and modify their affective state via INS activation, thereby perpetuating a cycle of addiction [51]. 

Few studies have utilized cold pressor tasks to examine pain sensitivity among individuals who use cannabis and findings have been mixed. For example, differences were not observed between individuals who reported frequent cannabis use (three or more uses per week) and those who reported no cannabis use for pain sensitivity, tolerance, or intensity in response to a CPT [53]. In contrast, Cooper and Haney [54] reported sex-based differences in pain sensitivity and tolerance using a CPT. Cannabis use was associated with decreased pain sensitivity among men but not women. However, increased pain tolerance was observed for both men and women immediately following cannabis use [54]. Importantly, neither of these studies reported diagnostic data regarding whether participants met criteria for CUD but instead focused on regular cannabis users; therefore, findings may differ among iCUD.

### 3.2. Conclusions

The reviewed literature provides strong neuroimaging evidence for altered brain processing among iSUD in brain regions implicated in interoception including INS, ACC, and thalamus. It is important to note that other interpretations of these findings are possible given the additional roles of INS and ACC in other processes including attention salience and conflict/error detection, respectively [55,56]. However, the findings reported above demonstrate associations between altered brain activity in INS/ACC in response to interoceptive stimuli and reduced indicators of IA suggesting a role for interoception in SUD. Disentangling these processes is a difficult endeavor and more research utilizing interoceptive perturbations is needed to extend these findings. 

In conjunction with heightened bottom-up interoceptive processing, there also appears to be evidence for decreased top-down cognitive control processes as illustrated by alterations in frontal regions including IFG and OFC. Behavioral tasks, such as heartbeat detection and cold pressor paradigms, have demonstrated mixed findings for altered responsivity to interoceptive stimuli across drug classes. However, some evidence has been presented to suggest an association between an aversive physical stimulus (cold-pressor) and increased craving and subsequent substance use among iAUD specifically. Taken together, findings suggest that interventions modifying activation within the above-mentioned brain regions may be effective for treating SUD. Mindfulness-based interventions are one such promising avenue for improving disrupted interoceptive processing in iSUD. Below, evidence for altering brain functioning with mindfulness-based practices is outlined and existing studies applying such interventions to treatment for iSUD are reviewed. 

## 4. Mindfulness

Over the past 30 years, the scientific community has demonstrated an increasing interest in the Buddhist-founded philosophy and practice of mindfulness. Mindfulness is defined as “the awareness that emerges through paying attention on purpose, in the present moment, and non-judgmentally to the unfolding of experiences moment by moment” [57]. It is considered an attention-training technique [58] with a focus on acceptance of internal and external signals of the body (e.g., sensations on the skin, interoceptive cues, and auditory signals) and the mind (e.g., thoughts and fluctuating emotions) [59]. Mindfulness, with its ability to teach people to adjust their awareness and ultimately reduce their stress, has potential for improving mental health, pain management, and SUD treatments [9,60,61].

### 4.1. Mindfulness-Based Interventions (MBI)

Several mindfulness-based interventions (MBI) have been developed including mindfulness-based stress reduction (MBSR), mindfulness-based cognitive therapy (MBCT), mindfulness-oriented recovery enhancement (MORE), and mindfulness-based relapse prevention (MBRP). The latter two, MORE and MBRP, were created specifically for iSUD, however, MBSR and MBCT have been used in the treatment of SUD as well.

Originally developed by Dr. Jon Kabat-Zinn, MBSR was created with the intention of improving quality of life for individuals with mental and physical health challenges as well as typical life stressors [62]. The focus of MBSR is to cultivate non-judgmental present moment awareness for the promotion of psychological and emotional resilience [62]. The ability to focus and refocus on these experiences, rather than allow for a wandering mind or implementing active problem solving, has been demonstrated to be very efficacious in many diverse populations [63]. Among iSUD, MBSR has been found to significantly improve overall quality of life by reducing distress in those with an extensive history of chronic substance use [58,63,64].

MBCT was created as a derivative from MBSR [65]. MBCT focuses on helping patients learn to facilitate awareness of their “greater freedom and choice” over their natural, habitual, and “overlearned automatic patterns of cognitive-affective processing” of complex thoughts and emotions [65]. This therapy was originally created to treat depression and the recurrence of depression by helping patients develop skills to disengage their thoughts, feelings, and bodily sensations from “automatic,” dysfunctional thinking patterns and enhance their ability to distinguish signs of a depressive episode in an effort to reduce the chance of relapsing [65]. Ultimately, MBCT plays a prominent role in mood/anxiety disorder research, which has, in turn, contributed the development of MBIs specific for iSUD. 

MORE was created as a blend of MBSR and MBCT practices with a meditative focus to aid in coping with drug cravings [66]. MORE also provides training and education for identifying processes of the mind (e.g., aversion, attachment, and thought suppression) as well as learning how to change or “mindfully let be” (i.e., accepting unchangeable circumstances) [66]. This training was created specifically for SUD symptoms, where the skillset of reappraisal (i.e., adjustment of one’s perspective or consideration of maladaptive thoughts that contribute to negative emotions and behaviors) and savoring (i.e., enjoying naturally rewarding experiences) can create a reduction in the valuing of substances, thus disrupting the cycle of craving [67]. 

Lastly, MBRP aims to help individuals increase present-moment awareness and increase self-efficacy [68,69]. This is accomplished by helping iSUD: (1) increase awareness of internal and external triggers that motivate substance use, including craving and negative affect; and (2) cultivate coping skills to tolerate these uncomfortable sensations without engaging in the habitual reaction of using substances [69,70]. With practice, iSUD develop the ability to recognize urges to use and make the conscious decision to not do so, instead developing alterative behavioral responses [69].

### 4.2. Brain Mechanisms of Mindfulness Practice

There is a growing body of literature demonstrating the effects of mindfulness and meditation practice on brain functioning among healthy individuals. These results generally suggest that the practice of mindfulness is associated with functional changes in interoceptive, and cognitive control brain regions shown to be altered within iSUD [70]. A recent review of longitudinal fMRI studies of mindfulness-based interventions identified increased INS activation as the most consistent and enduring finding across studies [71]. Preliminary evidence for an association between alterations in ACC activity and acceptance was also found. Despite this central finding, results across studies have been mixed. For example, decreased activation within INS, ACC, PFC, and precuneus was observed during active meditation among practitioners with a minimum of 4 years of experience [72] while Hölzel and colleagues [73] found greater activity in PFC and ACC during meditation among seasoned practitioners (8 years average experience) compared to novices, perhaps suggesting that amount of experience may impact findings. In addition to duration of mindfulness experience, variations in other sample characteristics and methodological differences, such as heterogeneity in the behavioral task used and an overall lack of prospective studies, may also contribute to the mixed findings observed between studies, pointing to the need for greater consistency in future study design.

Sustained changes in functional connectivity have also been found following an 8-week course in meditation with reductions in connectivity between INS and medial PFC (a region of the default mode network involved in repetitive negative thinking), and increased activation in dorsolateral PFC (involved in goal-directed activity and working memory updating); in contrast, non-meditating CTL showed strong connectivity between INS and medial PFC [74]. These findings indicate that meditation training may enhance the relationship between attentional focus and adaptive motivated behavior.

Associations between mindfulness and altered brain activity in response to negative emotional and interoceptive stimuli have also been demonstrated with and without formal mindfulness training. United States Marines who underwent a 20 h mindfulness-based training exhibited attenuated INS and ACC activity while experiencing an aversive interoceptive breathing load compared to baseline [75]. In contrast, there is also evidence to suggest that even present-moment mindfulness in novices may directly affect brain functioning; participants instructed to pay attention to the present moment non-judgmentally while viewing emotional images showed increased PFC activity when anticipating potentially negative images and reduced amygdala activity while viewing the images [76]. Individual differences in trait mindfulness were also linked with brain responsivity to emotional stimuli, such that INS activity while anticipating negative images positively correlated with self-reported trait mindfulness [76]. Similarly, individuals reporting high levels of dispositional mindfulness demonstrated a negative association between PFC and amygdala activation during an affect labeling task compared to those who reported low levels [77], perhaps suggesting that mindfulness is associated with greater cognitive control over emotional responsivity. Taken together, research thus far demonstrates a link between mindfulness/meditation and brain responsivity in brain regions associated with interoception and adaptive, goal-directed behavior among non-substance-using individuals, suggesting mindfulness interventions as a promising treatment option for SUD. 

### 4.3. Effects of MBI on Brain Regions Altered among iSUD

Research indicates that mindfulness-based interventions for SUD lead to significantly fewer days of substance use, reduced craving, and lower risk of relapse [70,78]. However, despite evidence from research with healthy CTL suggesting that mindfulness interventions influence brain regions altered among iSUD, studies using fMRI to examine the neural effects of mindfulness-based interventions in the treatment of SUD remain scarce. Thus far, neuroimaging research has predominantly been employed to examine mindfulness for the treatment of nicotine use. Westbrook and colleagues investigated the use of active mindful attention to reduce craving while viewing smoking cues among individuals who regularly use nicotine and had no formal meditation experience. Results demonstrated reductions in self-reported craving and activation in ACC as well as decreased functional connectivity between ACC and striatum, and INS, suggesting a decoupling of regions implicated in the interoceptive representation of craving [79]. Similarly, following 10 weeks of MORE treatment, individuals who used nicotine exhibited decreased ACC and striatum activation and increased functional connectivity between ACC and OFC compared to a nicotine-smoking control group who did not receive the MBI [67]. Reduced brain reactivity to stressful scenarios within INS, thalamus, amygdala, and other regions has also been observed in individuals receiving a mindfulness training compared to a cognitive behavioral therapy intervention for nicotine use [80].

To the best of our knowledge, neuroimaging techniques have only been utilized following MBI among iSUD in one other study. In that study, iOUD were assigned to receive either 4 weeks of a MBI in addition to treatment as usual (TAU) or TAU only during their first month at an addiction inpatient unit. Patients in the MBI group demonstrated a significant difference in resting-state functional connectivity within the default mode network, showing decreased connections with IFG post-MBI; this reduced IFG-default network connection was linked to greater self-reported mindfulness [81]. Other studies have also demonstrated that MBI can improve self-reported interoceptive awareness but do not examine brain correlates of these changes. For example, significant improvements in self-reported IA immediately following an adjunct Mindful Awareness in Body-oriented Therapy (MABT), as well as a relationship between sustained increases in IA and reduced risk for relapse after 12 months, have been demonstrated among women enrolled in an intensive outpatient program for chemical dependency [82,83]. Research thus far supports MBI as a promising treatment avenue for individuals who use nicotine. Given the evidence of altered interoceptive processing among individuals with various SUD, not only nicotine use disorder, it follows that MBI may be an effective treatment for other SUD populations as well. Substantial work is needed to test the efficacy of MBI for the treatment of other SUD and examination of the brain mechanisms of treatment effects should be a priority. Suggestions for future research are detailed below.

## 5. Proposed Future Directions

Mindfulness interventions hold significant promise in improving and maintaining recovery among iSUD. Specifically, by targeting, affective, cognitive, and interoceptive processes, mindfulness interventions may exert significant changes in reward-based learning, experience of negative affect, and craving and reactivity to drug use, all of which are associated with the emergence and maintenance of SUD. However, experimental data not only on the effectiveness of mindfulness-based interventions on SUD, but also the mechanisms and neurocircuitry that underlie it, are limited, and this warrants significant ongoing investigations. 

First, research is needed to establish that mindfulness-based interventions are efficacious across different substance classes. These efforts would benefit from including not only clinically diverse samples to take the often-observed comorbidity in this population into account, but also the moderating effect of demographic and social background information. Second, because several mindfulness training approaches exist, which practices, at what dosing, and in combination with what other interventions (e.g., cognitive restructuring, pharmacotherapy) lead to greater improvements needs to be established. For example, what are the difference in efficacy of mindfulness techniques employing focused attention on physical sensations relative to those focused on non-judgmental acceptance? Third, it will be important to employ multiple levels of analysis, including behavioral, physiological, and neurocircuitry. Fourth, based on previous investigations, the field would benefit from centering interoceptive processing as one of the key disease modifying mechanisms for iSUD. Such studies would be able to establish changes in interoception across levels of analysis, as well as the relationship to clinical and functional outcomes. Specifically, does mindfulness practice change interoceptive awareness and associated neural mechanisms, and does this change in turn mediate outcomes? Such studies will not only give evidence to the usefulness of mindfulness-based intervention in engaging underlying mechanisms in treatment of iSUD, but also identify potential predictors of outcomes, and thus additional targets for optimization of this intervention. Finally, engagement of these targets with augmentation approaches, such as biofeedback, real-time neurofeedback, or transcranial magnetic stimulation will provide evidence with respect to malleability of mechanisms underlying mindfulness, interoception, and SUD, as well as whether such augmentation may be necessary in more severe clinical presentations.

## 6. Conclusions

The literature reviewed here suggest that disruptions in brains’ interoceptive networks underlie the emergence and maintenance of SUD. Given that mindfulness practice engages similar networks, mindfulness-based interventions are well-poised to exert beneficial outcomes in iSUD on both neural and behavioral level. Indeed, mindfulness-based interventions have been associated with reductions in drug use and craving. In clinical settings, mindfulness may be effective in improving self-awareness and decreasing avoidance of inner states such as bodily sensations, emotions, and thoughts. Research shows that this might be beneficial for the often experienced, drug-use promoting negative affect (e.g., anxiety, distress, sadness) in iSUD, which can not only be present due to SUD comorbidities, but also directly result from drug withdrawal. This process may also translate to a reduction in responses to drug craving for iSUD. By learning to observe drug-cued mental and physical events, delaying responses to them, and substituting them with alternative behavioral responses, mindfulness may essentially teach iSUD to allow discomforting drug-related states to arise and dissipate naturally without the need for the drug’s potent effects. This disruption of the drug use cycle may then become self-reinforcing. On the other hand, mindfulness practice may bring about increases in experience of pleasure in non-drug related activities. By learning to be present in, notice, and savor meaningful, positive events, iSUD may come to increasingly approach such events in their daily lives. Nevertheless, research on the clinical application of mindfulness for SUD is in its early stages. Future research will need to identify whether mindfulness-based treatments are efficacious across substance classes, and what particular approaches and dosages show the largest effect sizes in enhancing neural networks disrupted by chronic substance use to improve short-term and long-term outcomes.

## Data Availability

Not applicable.

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
