# Peer review of "Mindfulness-Based Interventions for the Treatment of Aberrant Interoceptive Processing in Substance Use Disorders"

_brainsci, 2022, doi:10.3390/brainsci12020279_

Round 1

Reviewer 1 Report

The topic of the manuscript i.e. substance use disorders (SUD) is very important to the both public health and scientific community. The aim of this review is novel and interesting - the search for the evidence of interoceptive dysfunction in SUD and potential use of mindfulness meditation in this pathology . In my opinion the paper is scientifically accurate and it fits well the scope of the MDPI Brain Sciences. The title is long but informative. Abstract is informative and precise.  Chapters are not too long.

The language of the paper is scientific but clear and understandable. The number of analysed manuscripts and the method of the review are appropriate.  Each main chapter end with short conclusions. Authors added future directions for futher research in this field including the recovery from SUD with the use of Mindfulness meditation. The reference section is adequate and contains only the newest and most important citations.

Author Response

The authors thank the reviewer for their comments on our manuscript; we appreciate their description of the strengths of our paper. We agree that understanding the effects of mindfulness-based interventions for the treatment of SUD is important from both a public health perspective and for the scientific community and we are delighted for the potential opportunity to publish on such a critical topic.

Reviewer 2 Report

The current MS represents an elaborate and scholarly written narrative review, integrating and summarizing research on the relationship between mindfulness based interventions and anomalous processing of bodily signals (interoceptive processing) in Substance Use Disorder (SUD). This main goal is subdivided in three foci: the relationship between interoceptive processing and SUD, the relationship between mindfulness training and consequential brain activity changes, and mindfulness based trainings for SUDs.  The main preliminary conclusion is that SUD seems associated with heightened INS and ACC activity subsequent to drug cues, as well as blunted interoceptive processing related INS and ACC processing. And that mindfulness seems to reverse the aforementioned reduced INS and ACC processing. The manuscript is a well-structured, yet an elaborate read. However, there are some issues that deserve attention as listed below.

- It is very good that potential directions for future research are mentioned in the abstract. However, noting that SUD is characterized by enhanced INS and ACC responses to drug-cues, I think future studies should also focus on the effect of mindfulness approaches on reducing these responses to drug cues. As it stands, it is only mentioned in the abstract that studies should focus on the effect of enhancing INS and ACC to non-drug stimuli, yet I think the aforementioned focus may be at least (or perhaps more) relevant.

- In the abstract it is stated that mindfulness interventions in individuals with nicotine addiction are associated with increased INS/ACC responses, but this is not evident from section 4.3.

- In the introduction it is stated that the review focuses (in part) on the relationship between mindfulness and brain function in individuals not afflicted with SUD. That is of course still relevant in relation to SUD, but the relevance could be made clear(er).

- Page 2, line 71-72, it is stated that: “Alterations in this neural network, as reflected by reduced interoceptive accuracy (IA),[…]”. I think it would be helpful to the reader IA is (more elaborately) introduced here. What exactly is meant here by interoceptive accuracy and how is it generally operationalized (what reflects the reduced interoceptive accuracy)?.

- Throughout the manuscript there is a focus on the INS and ACC, that are thought to drive interoceptive processing. Certainly, it is well argued that these are key regions for interoceptive processing, yet these regions are also central to other processes such as conflict/error detection (ACC). Perhaps the authors could apply more nuance, or make more compelling argumentation in the discussion that the relationship between anomalous INS/ACC functioning in SUD is directly related to anomalous interoceptive processing, and not other processes (such as attention, error monitoring etc).

- As mentioned, conflicting findings have been found regarding the effect of mindfulness training on INS responsivity (e.g. section 4.2.), how can these conflicting findings be explained? I think that is important to elaborate, especially in view of the conclusion that states that mindfulness is associated with an increase in INS and ACC processing related to interoceptive processing.

- Two main relationships are the blunted INS/ACC response in relation to interoceptive processing and heightened INS/ACC processing in relation to drug-cues. Though there is much emphasis on the first relationship, the manuscript may benefit from more elaboration regarding the functional importance of the latter relationship. In other words, how does the relationship between INS/ACC processing in response to drug-cues relate to the behavioural problems in iSUD, and what is its relevance in relation to interoceptive processing?

- Though clear future directions are stated, I think the MS would benefit from a short conclusion that includes potential clinical implications.

- Typo in title: Abberant. Should be Aberrant.

Round 2

Reviewer 2 Report

The revised manuscript is improved significantly, and all my comments have been addressed.